# Pushing thermal conductivity to its lower limit in crystals with simple structures

Zezhu Zeng [1,2,6] ✉, Xingchen Shen [3,6], Ruihuan Cheng [1], Olivier Perez [3], Niuchang Ouyang[1], Zheyong Fan [4], Pierric Lemoine [5], Bernard Raveau[3], Emmanuel Guilmeau [3] ✉ & Yue Chen [1] ✉

Materials with low thermal conductivity usually have complex crystal structures. Herein we experimentally find that a simple crystal structure material AgTlI$_2$ (I4/mcm) owns an extremely low thermal conductivity of 0.25 W/mK at room temperature. To understand this anomaly, we perform in-depth theoretical studies based on ab initio molecular dynamics simulations and anharmonic lattice dynamics. We find that the unique atomic arrangement and weak chemical bonding provide a permissive environment for strong oscillations of Ag atoms, leading to a considerable rattling behaviour and giant lattice anharmonicity. This feature is also verified by the experimental probability density function refinement of single-crystal diffraction. The particularly strong anharmonicity breaks down the conventional phonon gas model, giving rise to non-negligible wavelike phonon behaviours in AgTlI$_2$ at 300 K. Intriguingly, unlike many strongly anharmonic materials where a small propagative thermal conductivity is often accompanied by a large diffusive thermal conductivity, we find an unusual coexistence of ultralow propagative and diffusive thermal conductivities in AgTlI$_2$ based on the thermal transport unified theory. This study underscores the potential of simple crystal structures in achieving low thermal conductivity and encourages further experimental research to enrich the family of materials with ultralow thermal conductivity.

Pushing the lattice thermal conductivity ($\kappa$) of materials to a lower limit[1,2] has attracted extensive attention in both condensed matter physics and materials science[3,4] as it is crucial to promote applications of thermal insulation and thermoelectrics[5,6]. Phonon, as a collective thermal excitation, plays a crucial role in the heat transport of solid crystals. Therefore, effectively impeding the phonon transport can significantly decrease the $\kappa$ of solids.

There are two main mechanisms to scatter phonons in semiconductors and insulators: intrinsic phonon-phonon interactions and extrinsic phonon scatterings from phonon-impurity or phonon-boundary interactions. Significant intrinsic phonon-phonon scatterings usually exist in materials with complex crystal structure[7,8] and strong lattice anharmonicity[9]. Examples of low $\kappa$ materials at room temperature include Cu$_{12}$Sb$_4$S$_{13}$ (0.69 W/mK)[7] and Cs$_2$PbI$_2$Cl$_2$ (0.45 W/mK)[10]. However, the unified theory developed by Simoncelli et al.[11–15] suggests that complicated crystal structure inevitably introduces considerable non-diagonal $\kappa$ originated from the wavelike tunnelling of adjacent phonons, which noticeably enhances the overall $\kappa$ of materials and indirectly impedes the progress of $\kappa$ toward its lower limit.

[1]Department of Mechanical Engineering, The University of Hong Kong, Pokfulam Road, Hong Kong SAR, China. [2]The Institute of Science and Technology Austria, Am Campus 1, Klosterneuburg, Austria. [3]CRISMAT, CNRS, Normandie Univ, ENSICAEN, UNICAEN, Caen, France. [4]College of Physical Science and Technology, Bohai University, Jinzhou, China. [5]Institut Jean Lamour, UMR 7198 CNRS - Université de Lorraine, Nancy, France. [6]These authors contributed equally: Zezhu Zeng, Xingchen Shen. ✉e-mail: zzeng@ist.ac.at; emmanuel.guilmeau@ensicaen.fr; yuechen@hku.hk

Extrinsic phonon scatterings from lattice defects, boundaries, and distortions provide additional ways to scatter phonons efficiently. Superlattice $Bi_4O_4SeCl_2$[16] creatively achieves the lowest $\kappa$ of 0.1 W/mK at 300 K along the out-of-plane direction in all bulk inorganic solids, via artificially manufactured orientation-dependent lattice distortion (layered superlattice structure) and the introduction of defects (a 88% density of the theoretical value). However, substantial defects in crystals can influence material properties such as electron mobility, toughness, and strength[17]. Moreover, directional manipulation of bond strength and connectivity in $Bi_4O_4SeCl_2$ can only affect the heat transfer in the corresponding direction, and its in-plane $\kappa$ (1.0 W/mK at 300 K) is still relatively high.

These aforementioned limitations greatly hinder the development of new materials with ultralow $\kappa$. In this case, exploring simple crystal structure compounds (low non-diagonal $\kappa$) with intrinsically ultralow (no artificial modification) $\kappa$ can be an alternative solution. It has been reported that PbTe (2 W/mK)[18], InTe (1.0 W/mK)[19], TlSe (0.5 W/mK)[20], $Tl_3VSe_4$ (0.30 W/mK)[21] possess low $\kappa$ at 300 K, which open the door to the discovery of ultralow $\kappa$ in simple crystal structures.

Herein, we synthesized a simple crystal structure phase $AgTlI_2$ (space group I4/mcm; eight atoms in the primitive cell (PC)) and found that it exhibits an exceedingly low $\kappa$ of 0.25 W/mK at room temperature. $AgTlI_2$ was synthesized[22,23] and found to be an electrical insulator[23], while no measurement of $\kappa$ has been reported in the literature. We performed single-crystal diffraction refinement and found a considerable negative part of the probability density function for Ag atoms, directly unearthing the strongly anharmonic nature of Ag atoms. Based on the ab initio molecular dynamics (AIMD) trajectories at 300 K, a large atomic displacement parameter (ADP) and rattling behaviour of Ag atoms are also revealed. These unusual phenomena imply strong lattice anharmonicity in $AgTlI_2$. Using unified theory[13] within the first-principles calculations, we provide further insight into the ultralow $\kappa$. In our calculations, a combination of factors is considered, including lattice thermal expansion, temperature-dependent anharmonic force constants[24], phonon frequency renormalization[25,26] and four[27,28] phonon scatterings. Our results show that there is a coexistence of suppressed propagative and diffusive heat transfer, uncovering the nature of the ultralow $\kappa$ in $AgTlI_2$. Based on an in-depth understanding of the heat transport in $AgTlI_2$, we also propose a potential alternative approach to push $\kappa$ to its lower limit.

## Results and discussion
### Experiments
$AgTlI_2$ and $AgTl_2I_3$ both belong to AgI-TlI quasi-binary silver-thallium iodide system, and they constitute an adjacent regime of the Ag-Tl-I equilibrium phase diagram[22], which inevitably leads to the formation of these two phases during material synthesis. Here, we conducted a low-temperature long-time annealing process in sealed tubes at 373 K to synthesize high-purity sample (see 'Methods' section). Rietveld refinement of the powder X-ray diffraction pattern (PXRD) of the as-synthesized powder (see Fig. S1 of Supplementary Information (SI)) confirms the high purity of our sample, with only traces of $AgTl_2I_3$.

The crystal structure of $AgTlI_2$ was established by Flahaut et al.[29]. We performed single-crystal diffraction refinement (see 'Methods' for the preparation and characterization) and we confirm that tetragonal crystal structure of this iodide (Fig. 1a, b) has a strong one-dimensional character. It consists of infinite $[AgI_2]^-$ chains of edge-sharing $AgI_4$ tetrahedrons running along $c$-axis. These anionic chains are isolated one from the other and the cohesion of the structure is ensured by the presence of $Tl^+$ cations between them (Fig. 1a, b). We demonstrate from this single-crystal study that $Ag^+$ exhibits a strongly anisotropic vibration inside "I4" tetrahedron. The latter cation is characterized by a maximum quadratic displacement of 0.056 $Å^2$, corresponding to a maximum vibration of 0.24 Å around its equilibrium position. Importantly, the refined residual electron density around the Ag site in Fig. 1c using harmonic description

(see 'Methods' for the details) reveals the presence of significant negative and positive parts of residues around the silver atom, suggesting the invalidation of the harmonic description for Ag atoms. In Fig. 1c, the absence of the negative part of the probability density function (pdf) of Ag atoms by the introduction of third-order Gram-Charlier anharmonic atomic displacement parameter (ADP) (see 'Methods' for details), further indicates the largely anharmonic nature of Ag atoms. The refined crystallographic data and anisotropic ADP ($U_{ii}$) of the single-crystal $AgTlI_2$ at 300 K can be found in Table S1 and S2, respectively. In contrast to Ag, Tl and I elements exhibit distinctly lower ADP. While Tl can also be regarded as a potential rattler in $AgTlI_2$ as the ADP of Tl is comparable to the corresponding values observed in many reported materials that exhibit Tl-rattling behaviour (see Fig. S5 for comparison).

In order to experimentally determine the thermal conductivity, the as-synthesized powder was densified by Spark Plasma Sintering (SPS, see 'Methods' section for details). Rietveld refinement of the PXRD pattern of the SPS-ed powder (see Fig. S2) confirms that the tetragonal crystal structure of the compound and the high purity of the sample are retained after sintering. Only traces of $AgTl_2I_3$ with a minor percentage (<1%) is detected. The refined crystallographic data of $AgTlI_2$ and $AgTl_2I_3$ phases are shown in Table S3 of SI. This minor secondary phase makes a negligible contribution of $\kappa$ to the bulk sample. The temperature ($T$) dependence of the thermal conductivity from 4 to 325 K, displayed in Fig. 1d, shows an extremely low $\kappa$ of 0.25 W/mK at 300 K and features a crystalline peak of $\kappa$ at an ultralow temperature of 8 K. Moreover, $AgTlI_2$ sample follows an unconventional and weak temperature-dependent $T^{-0.21}$ relation of $\kappa$ from 100 to 325 K, indicating an anomalous lattice dynamics and thermal transport mechanism. A comparison of the thermal conductivity of $AgTlI_2$ with representative materials featuring simple crystal structures (Fig. 1d) further reveals that $AgTlI_2$ exhibits at room temperature one of the lowest thermal conductivity among them.

### Molecular dynamics simulations
Bearing in mind from previous studies of low $\kappa$ compounds, such as TlSe[20] and $Tl_3VSe_4$[21], that Tl atoms usually exhibit highly anharmonic vibrations, we perform AIMD simulations using the VASP package[30,31] to investigate the atomic vibrations. The significant anisotropic atomic displacement parameters (ADPs) $U_{ii}$ ($i = 1, 2, 3$), displayed in Fig. S7, from 100 to 300 K based on the AIMD atomic trajectories illustrated in Fig. S6, reveal the magnitude of atomic thermal motion. Strikingly, we find that the ADPs of Ag atoms are significant with nonlinear enhancement from 100 K to 300 K. The magnitudes of $U_{ii}$ for Tl and I atoms are also smaller than that of Ag atoms with a much linear thermal evolution. The results confirm the single-crystal diffraction experimental data and reveal the strong anharmonicity of Ag ions in $AgTlI_2$. Previous studies have attributed the low $\kappa$ in TlSe[20] and $Tl_3VSe_4$[21] to highly anharmonic vibrations of Tl atoms. In $AgTlI_2$, Ag atoms show larger oscillations than Tl at 300 K, which indicates the ultralow $\kappa$ of $AgTlI_2$ may be more related to Ag element.

In Figure 2a, we compare the maximum experimental $U_{ii}$ values between Ag element in $AgTlI_2$ and Cu and Ag elements (typical rattlers) in thermoelectric materials with ultralow $\kappa$ at 300 K. The comparison clearly illustrates that the oscillation of Ag atoms in $AgTlI_2$ surpasses that of most rattling atoms in typical low $\kappa$ materials. The $U_{max}$ in $AgTlI_2$ is even comparable to the $U_{max}$ of Cu in three-fold coordination encountered in tetrahedrite $Cu_{12}Sb_4S_{13}$[32]. These unique features further highlight the strong lattice anharmonicity of $AgTlI_2$.

Knoop et al. proposed a parameter ($\sigma^A$)[33,34] to quantitatively evaluate the degree of lattice anharmonicity as

$$\sigma^A(T) \equiv \frac{\sigma[F^A]_T}{\sigma[F]_T} = \sqrt{\frac{\sum_{i,\alpha}\left\langle \left(F_{i,\alpha}^A\right)^2 \right\rangle_T}{\sum_{i,\alpha}\left\langle \left(F_{i,\alpha}\right)^2 \right\rangle_T}}, \tag{1}$$

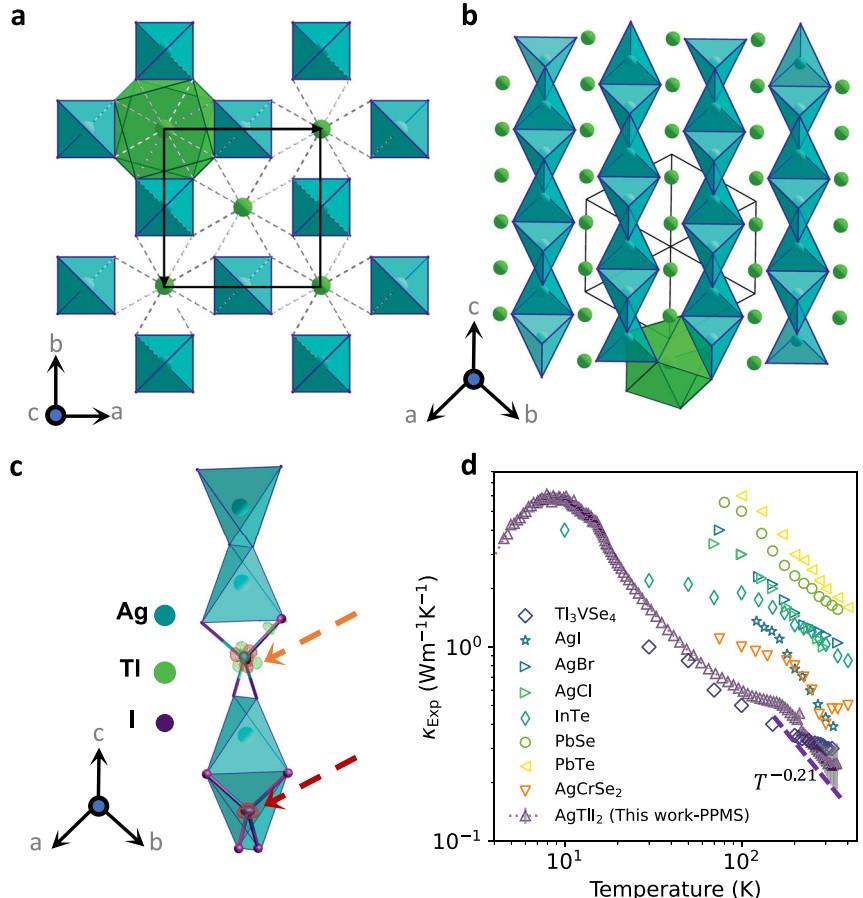

**Fig. 1 | Structure analysis and experimental thermal conductivity of AgTlI₂.**
Structural projection **a** along *c*-axis, **b** along [111] and focus on a [AgI₂]∞ chain. Ag, Tl, and I atoms are drawn using blue, green, and purple circles, respectively. **c** Orange arrow: Residual electron density (red: positive and green: negative) in the area of the Ag site for a harmonic description of Ag. Red arrow: Anharmonic three-dimensional probability density function (pdf) isosurfaces of Ag (red cloud); displacements are pointing toward the faces of the tetrahedron. **d** Experimental thermal conductivity of AgTlI₂ from 4 to 325 K measured by Physical Property Measurement System (PPMS). Data for Tl₃VSe₄[21], AgI[56], AgBr[57], AgCl[58], InTe[19], PbSe[59], PbTe[59], and AgCrSe₂[60,61] were obtained from previous experiments.

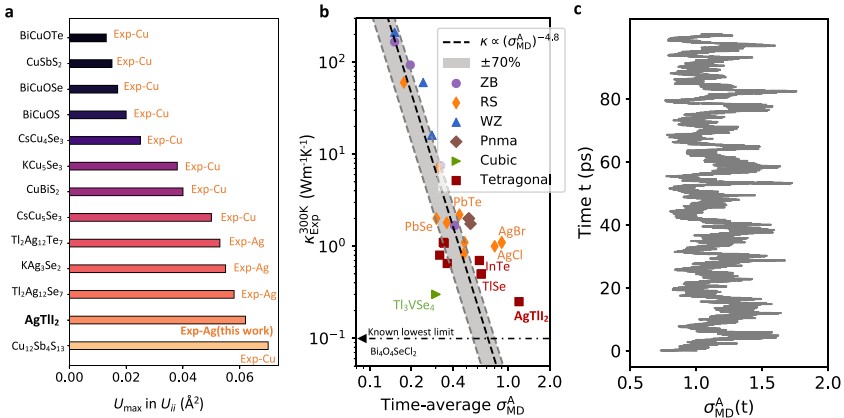

**Fig. 2 | Evaluation of atomic thermal motions and anharmonicity of AgTlI₂.**
**a** The maximum ADP (U_max) at 300 K in experiments for rattling Cu/Ag atoms in AgTlI₂ and typical thermoelectric materials[62–69]. **b** Relation between anharmonic parameter ($\sigma^A$)[33,34] and experimental $\kappa$[33,34,70] of typical simple crystals at 300 K. The data can be found in Table S4 of Supplementary Information. **c** Time-dependent anharmonic parameter $\sigma^A$ of AgTlI₂ calculated from AIMD simulations at 300 K.

where $i$, $\alpha$ and $T$ are, respectively, the atomic index, Cartesian direction, and temperature. $F^A$ and $F$ represent the anharmonic and total atomic forces, respectively. Note that $\sigma^A$ is time-dependent in AIMD simulations, and we can derive its time average readily. For most materials with modest anharmonicity, it will converge rapidly with

the increase of simulation time[33,34]. We computed (see 'Methods' for details) the $\sigma^A$ of AgTlI₂ and some typical crystals (Tl₃VSe₄, TlSe, PbTe, PbSe, AgCl and AgBr) with simple crystal structures and ultralow $\kappa$ at 300 K from AIMD simulations, and make a comparison (Fig. 2b) with the $\kappa$-$\sigma^A$ power-law model proposed by Knoop et al.[33,34].

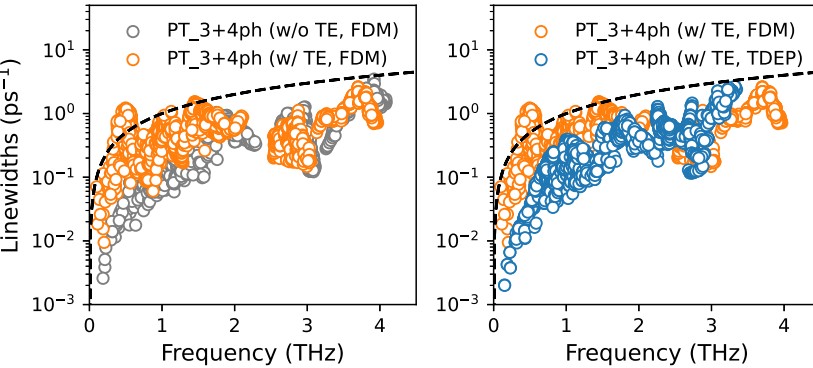

**Fig. 3 | Phonon linewidths of AgTlI$_2$ at 300 K.** Phonon linewidths (including three- and four-phonon scatterings) of AgTlI$_2$ at 300 K calculated using perturbation theory (PT) considering different phonon frequencies obtained via finite displacement method (FDM) and temperature-dependent effective potential (TDEP) scheme with or without the lattice thermal expansion (TE). The black dashed lines represent the phonon linewidths that are equal to the corresponding phonon frequencies.

All harmonic atomic forces are computed using the finite displacement method[35].

It is found that AgTlI$_2$ owns the largest time-average $\sigma^A$ among these materials and significantly deviates from the power-law model at 300 K, which are the intrinsic imprints of its large atomic displacement parameters and strong lattice anharmonicity, further implying its ultralow $\kappa$. We note that the calculated $U_{max}$ for Ag in AgTlI$_2$ at 300 K (0.097 Å$^2$) based on AIMD trajectories is larger than that of our experiments (0.061 Å$^2$), which may be used to rationalize the significant deviation of $\sigma^A$ from the power-law model. Moreover, the time evolution (Fig. 2c) of $\sigma^A$ of AgTlI$_2$ shows pronounced fluctuations in a period of several picoseconds, a behaviour not observed in other materials with similar crystal structures, such as InTe (see Fig. S8).

Knoop et al.[34] reported the hopping of $\sigma^A$ in CuI and KCaF$_3$, which can be ascribed to the formation of some metastable defect geometries. However, the intrinsic mechanism is different here; there is no metastable structure formed in AgTlI$_2$. A fluctuation of $\sigma^A$ was also reported by Knoop et al.[33] in rock-salt AgBr at 300 K, a well-known ionic conductor[36]. Fortunately, our experimental measurements do not indicate any superionic diffusion or ionic thermal conductivity in AgTlI$_2$, which contributes to the stability of an exceptionally low $\kappa$ in AgTlI$_2$.

## Anharmonic lattice dynamics

Atomic vibrations can be directly observed in MD simulation, while it is nontrivial to obtain more insights into the microscopic phonon transport mechanisms. Thus, we further study the anharmonic lattice dynamics within the perturbation theory (PT) on the basis of first-principles calculations. According to the unified theory[13], the lattice thermal conductivity can be written as

$$
\kappa = \frac{\hbar^2}{k_B T^2 V N_q} \sum_q \sum_{j,j'} \frac{\omega_q^j + \omega_q^{j'}}{2} \mathbf{v}_q^{j,j'} \otimes \mathbf{v}_q^{j',j}
$$
$$
\cdot \frac{\omega_q^j n_q^j (n_q^j + 1) + \omega_q^{j'} n_q^{j'} (n_q^{j'} + 1)}{4 (\omega_q^{j'} - \omega_q^j)^2 + (\Gamma_q^j + \Gamma_q^{j'})^2} (\Gamma_q^j + \Gamma_q^{j'}),
\tag{2}
$$

where $T$, $\hbar$, $k_B$, $V$ and $N_q$ are respectively the temperature, reduced Plank constant, the Boltzmann constant, the volume of the unit cell, and the total number of wave vectors. $\mathbf{v}_q^{j,j'}$, $\Gamma_q^j$ and $\omega_q^j$ are, respectively, group velocity, linewidth, and frequency of a specific phonon with wave vector $\mathbf{q}$ and branches $j$ and $j'$. When $j = j'$, Eq. (2) computes the conventional propagative thermal conductivity ($\kappa_{pg}$), and if $j \neq j'$, it calculates the diffusive thermal conductivity ($\kappa_{diff}$).

Equation (2) indicates that phonon frequency $\omega$ and linewidth $\Gamma$ are two crucial factors that determine $\kappa$. Therefore, a sophisticated treatment of these parameters is necessary for a strongly anharmonic system. We consider a combination of factors including: lattice thermal expansion, temperature-dependent phonon frequency renormalization[25,26,37] and four-phonon interactions[27,28]. We carefully compare the effects of these factors on the phonon frequencies (see Fig. S9) and linewidths (see Fig. 3) of AgTlI$_2$ at 300 K, and find that all of them are important to reliably understand the $\kappa$ of AgTlI$_2$. In Fig. 3, the dashed black lines compare phonon frequency and linewidth for vibrational modes at 300 K. The majority of vibrational modes exhibit well-defined phonons, with linewidths smaller than their corresponding frequencies. Therefore, the unified theory[13] using the Lorentzian spectral function approximation (Eq. (2)) used in this work is reliable to compute $\kappa$. However, in cases where overdamped phonons (non-Lorentzian phonon spectral function) dominate, a more sophisticated approach considering the full phonon spectral function integral in the Wigner[38] or Hardy's[39,40] heat flux space becomes necessary for reliably assessing $\kappa$.

In Figure 4a, b, $\kappa_{pg}$ and $\kappa_{diff}$ calculated using different combinations of phonon frequencies and linewidths are reported at 300 K. First, we find that lattice thermal expansion (TE) can largely decrease the $\kappa_{pg}$, as the TE significantly softens the low-frequency phonons and enhances the phonon scatterings (see Fig. S9 and Fig. 3). The enhancement of low-frequency phonon linewidths is strongly related to the stronger couplings of low-frequency acoustic and optical phonon modes as these softened phonons have a wider energy distribution and thus easier to satisfy the energy conservation after considering TE. Interestingly, the $\kappa_{pg}$ is enhanced markedly when phonon frequency renormalization is considered via the temperature-dependent effective potential (TDEP) scheme[25,26]. In fact, this phenomenon was also reported in previous studies[7,10] as the low-frequency phonons (main heat carriers) are hardened significantly (see Fig. S9) due to the high-order anharmonic renormalization and thus weakening of phonon interactions. By contrast, $\kappa_{diff}$ is not as sensitive as $\kappa_{pg}$ over TE and frequency renormalization. With these necessary considerations, the predicted overall thermal conductivity ($\kappa_{pg} + \kappa_{diff}$) agrees well with the experiments.

Comparing the spectral-$\kappa$ with respect to the phonon frequency as shown in (Fig. 4c, d), we see that the low-frequency phonons (0–1 THz) have significant contributions to the propagative thermal transport. From the phonon density of states at 300 K (see Fig. S10), we know that these phonons are mainly dominated by the vibrations of Tl atoms. The diffusive thermal transport can be greatly contributed by the phonons in the frequency ranges of 0.8–1.5 and 2–3 THz, and these

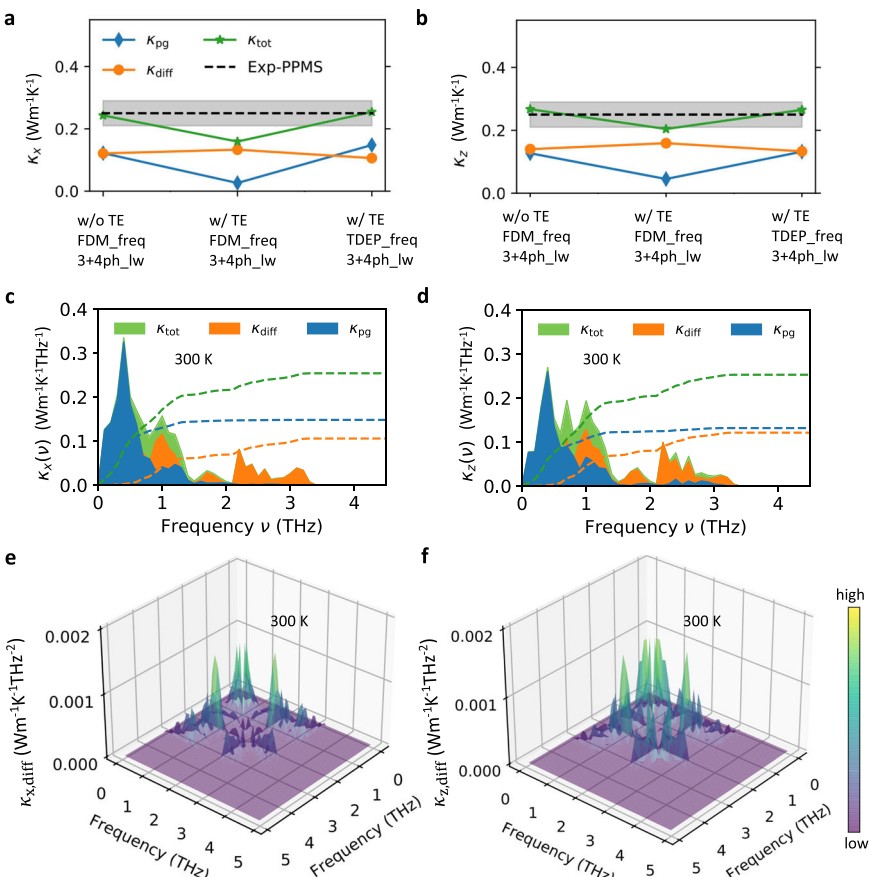

**Fig. 4 | Calculated lattice thermal conductivity of AgTlI$_2$ within unified theory at 300 K.** Lattice thermal conductivities of AgTlI$_2$ calculated at 300 K along $x$ (**a**) and $z$ (**b**) directions using three different sets of phonon properties extracted from perturbation theory. Solid lines are guide for eyes. TE represents thermal expansion. Phonon frequencies (freq) have been extracted from the finite displacement method (FDM)[35] and the temperature-dependent effective potential (TDEP) scheme[25,26]. For phonon linewidths (lw), three-[55] and four-phonon interactions[27,28] are included. Experimental results (black dashed line) with measurement uncertainty (grey shaded area) are shown. Two-channel spectral lattice thermal conductivities ($\kappa_{pg}$ and $\kappa_{diff}$) calculated using the unified theory[13] at 300 K (using TDEP phonon frequency and 3 + 4 ph_lw) along $x$ (**c**) and $z$ (**d**) directions, and the three-dimensional frequency-plane distribution of $\kappa_{diff}$ along $x$ (**e**) and $z$ (**f**) directions.

phonons are strongly related to Ag atoms as revealed by phonon density of states (see Fig. S10). Based on the three-dimensional distribution of $\kappa_{diff}$ as shown in (Fig. 4e, f), it is also apparent that wavelike tunnellings between low-frequency (-1 THz) and high-frequency phonon modes (2–3 THz) can considerably contribute to the diffusive thermal transport. Combined phonon density of states with spectral-$\kappa$, we conclude that the strong oscillation and anharmonicity of Ag atoms lead to two crucial thermal transport features in AgTlI$_2$: (i) dramatically scattered high-frequency phonons and effectively decreased $\kappa_{pg}$; (ii) glass-like thermal transport and non-negligible $\kappa_{diff}$. While the presence of Tl atoms, which dominate the vibration frequency range of 0.5–0.8 THz, contributes significantly to the scattering of low-frequency phonons and further decreases the $\kappa_{pg}$. With weak atomic bonding and simple crystal structure, AgTlI$_2$ avoids severe phonon bunching and achieves a coexistence of both ultralow $\kappa_{pg}$ and $\kappa_{diff}$ at 300 K.

**Pushing low $\kappa$ to a lower limit**
The discovery of ultralow-$\kappa$ materials at room temperature in simple and fully dense crystal structures has proven extremely challenging. In Fig. 5, we show a schematic diagram for potential different pathways to find lower $\kappa$. A current protocol (red dashed line) for finding ultralow-$\kappa$ materials is enhancing the complexity of materials. Along this line, some complex materials with low $\kappa$ were found readily such as skutterudite YbFe$_4$Sb$_{12}$[41] and perovskite CsPbBr$_3$[5], as complex materials normally possess lower crystal symmetries and stronger phonon couplings, and thus smaller $\kappa_{pg}$. However, under the framework of the two-channel thermal transport, this protocol may not be efficient to suppress $\kappa$ if there is a non-negligible diffusive heat transport channel ($\kappa_{diff}$). In contrast, we even see a higher $\kappa$ when the complexity of materials is further raised, such as tetrahedrite Cu$_{12}$Sb$_4$S$_{13}$[7] and argyrodite Ag$_8$GeS$_6$.

Ag$_8$GeS$_6$ (60 atoms in PC)[42] is an excellent example to explain this phenomenon. We compute its $\kappa_{pg}$ using unified theory and find that it is only 0.04 W/mK at 300 K, which is lower than that of AgTlI$_2$ and even close to the $\kappa$ of air (0.025 W/mK at 300 K). Nevertheless, the $\kappa_{diff}$ is equal to 0.43 W/mK and it is much higher than $\kappa_{pg}$, which can be attributed to the severe coherence of phonon modes (180 phonon branches in the first Brillouin zone) with strong lattice anharmonicity (see Fig. S13 for phonon properties and two-channel $\kappa$ of Ag$_8$GeS$_6$). In other words, although strong lattice anharmonicity in complex materials can dramatically decrease $\kappa_{pg}$ (purple arrows in Fig. 5), the total $\kappa$ can increase due to the high diffusive thermal transport. We highlight that the coexistence of ultralow $\kappa_{pg}$ and $\kappa_{diff}$ is the key to achieve strongly suppressed $\kappa$ in crystals. AgTlI$_2$ delivers on both fronts, demonstrating the potential of simple crystal structures with strong lattice anharmonicity to push $\kappa$ to a lower limit. Therefore, finding materials with simple crystal structures and giant lattice anharmonicity (quantified by $\sigma^A$[34]) can be a feasible way to push $\kappa$ to its lower limit.

Recent studies have focused on finding high thermoelectric $zT$ values from complex materials[43] and even high-entropy alloys[44]. However, an enhanced $\kappa$ at elevated temperature was observed in

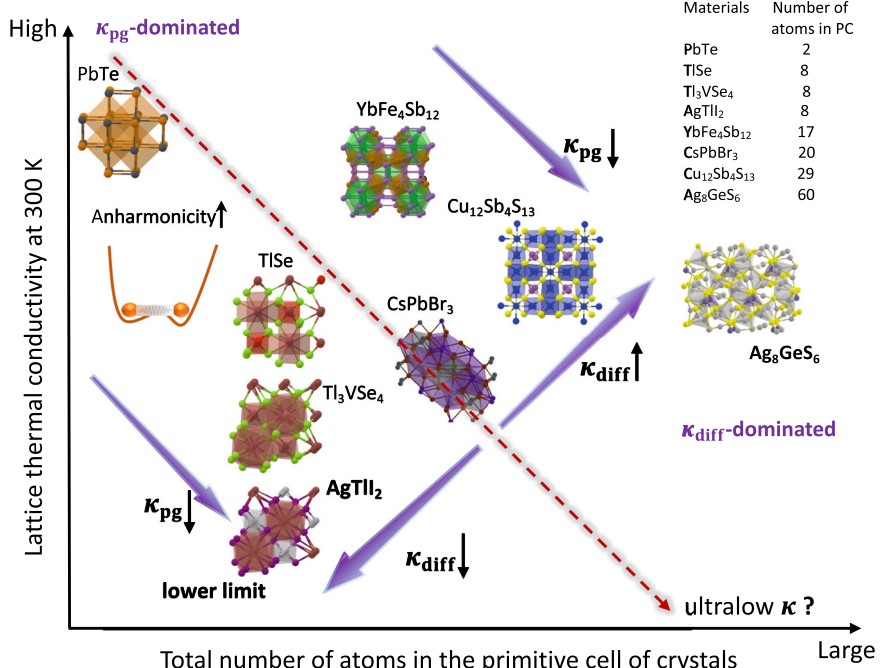

**Fig. 5 | Potential pathways for pushing $\kappa$ to its lower limit in inorganic materials.** The dashed red line represents the current protocol to find materials with lower thermal conductivity. Lattice thermal conductivities are extracted from previous studies based on unified theory[7,13,41,71]. We compute the two-channel $\kappa$ of AgTlI$_2$ and Ag$_8$GeS$_6$ in this work.

complex materials such as high-entropy alloy BiSbTe$_{1.5}$Se$_{1.5}$[44] and Zintl compound BaAg$_2$Te$_2$ (20 atoms in PC)[45]. In fact, these complex materials already show the glass-like heat transport behaviour around room temperature (large $\kappa_{diff}$), similar with Ag$_8$GeS$_6$. The increase of $\kappa$ considerably affects the efficiency of thermal energy conversion at high temperatures. Given that AgTlI$_2$ is an insulator with a large electronic band gap (Fig. S11), it is not suitable for direct thermoelectric applications without proper alloying or doping. Nonetheless, the microscopic insights obtained from studying its thermal transport behaviours, along with those in Ag$_8$GeS$_6$, contribute to a better understanding of heat transport in thermoelectric materials.

In summary, we synthesized polycrystalline AgTlI$_2$ with a simple crystal structure, and reported its ultralow $\kappa$ at room temperature via both state-of-the-art experiments and calculations. This unusually low value of $\kappa$ is close to the known lowest limit[16] of any bulk inorganic materials. We thoroughly investigate its thermal transport nature by revealing the large thermal vibrations of Ag atoms from single-crystal diffraction refinements and AIMD simulations, and uncover the origin of simultaneously suppressed particle-like and wave-like phonon thermal transports based on the unified theory. Compared with previous efforts in searching for ultralow $\kappa$ in complex materials, our work emphasizes the unique advantages and features of simple crystal structures, and also suggests an alternative and practicable way to find new inorganic materials with ultralow thermal conductivity.

## Methods

### Sample preparation
The nominal AgTlI$_2$ sample was prepared starting from the high-purity precursors AgI (powder, 99.999%) and TlI (ball, 99.999%). A stoichiometric amount of 3 g precursors was mixed and loaded into a carbon-coated silica tube inside of the glove-box, followed by sealing and evacuating (-10$^{-3}$ Pa) under vacuum. The mixture of precursors was melted at 873 K for 24 h, and then slowly dropped to 373 K over 24 h, followed by annealing at this temperature for 240 h. The obtained ingot was hand-ground into fine powders. Finally, they were densified by SPS at 373 K under a pressure of 200 MPa for 10 min using a 10 mm

diameter WC die. The resulting SPS-ed sample has a geometrical density of 6.85 g/cm$^3$, i.e., around 99% of the theoretical density.

### X-ray diffraction
PXRD of the synthesized powder, SPS-ed powder, and consolidated bulk sample were recorded on a PANalytical XPert2 system with Cu K$_{\alpha 1}$/K$_{\alpha 2}$ radiation ($\lambda_1$ = 1.5406 Å, $\lambda_2$ = 1.5443 Å). Rietveld refinements of PXRD patterns were performed using GSAS (General Structure Analysis System) packages[46]. In Fig. S3, XRD patterns of the bulk sample (surface perpendicular to the SPS pressure direction) have a comparable pattern as that of the SPS-ed powder sample, implying non-preferential orientation in the bulk sample.

The batch synthesis was composed of grains of matter a few tens of micrometres in size. These grains of matter were not single-crystals, and were broken up with a scalpel and selected using a stereomicroscope. They were mounted on microloops and their diffraction quality was tested. After numerous trials and errors, we finally succeeded in isolating a single crystal of the targeted phase of sufficient quality; the dimensions of this crystal were 0.011 × 0.008 × 0.008 mm$^3$. X-ray diffraction measurement on a single crystal was then performed on Rigaku Synergy S diffractometer, equipped with a micro-focus sealed X-ray Mo tube and an Eiger 1M Dectris photon counting detector. Due to the very small size of the sample, an extremely long exposure time was chosen (360 s/0.5°); the utilization of the measurement strategy enabled us to obtain 97% completeness at a theta angle of 36.9°. Data reduction was performed with CrysAlisPro. The structure determination was then performed using SHELXS3[47] in the I4/mcm space group and refined using Olex2[48]. Although the structural refinement was quite satisfactory, the Fourier difference map revealed the presence of significant residues around the silver atom. Four positive residues and four negative residues of equal weight surrounded the silver atom (Fig. 1c), indicating a poor description of this atom. Silver cation with d$^{10}$ configuration can easily adopt various complex asymmetric coordination; Ag$^+$ ions can be observed in different but overlapping sites. The structure is then characterized by the presence of static or dynamic disorders. Whatever the situation, the description of the site

is complex and the use of higher-order tensor elements to model the ADP[49] can be an elegant solution to this problem.

The Gram-Charlier formalism is recommended by the IUCr Commission on Crystallographic Nomenclature. Note that the anharmonic approach has been successfully used in the past to solve numerous structures, both of nonconducting materials[50] and of fast ion conducting phases[51]. In the present case, the introduction of third-order Gram-Charlier anharmonic ADP for the silver site significantly improved the refinement to R = 1.97% (against 3.44%) for one additional parameter, with a drop of the residuals in the Fourier difference maps ([$-0.7\,e^-/Å^3 - 1.8\,e^-/Å^3$] against ([$-2.5\,e^-/Å^3 - 2.7\,e^-/Å^3$]) in the harmonic approach. The probability density function (pdf) for the silver site is plotted in Fig. 1c; the absence of a negative part in the pdf validates the present model. As already reported by different authors for $d^{10}$ elements, the probability density deformation increases the electron density of the Ag site toward the faces of the tetrahedron.

## Thermal transport measurements

Thermal conductivities of $AgTlI_2$ bulk sample were measured from 4 to 325 K using a Physical Property Measurement System (PPMS, Quantum Design, Dynacool 9 T). In order to cross-check $\kappa$ of $AgTlI_2$ obtained from PPMS, we also measured thermal diffusivity and thermal conductivity (see Fig. S4) on a Netzsch LFA 457 laser flash system under a nitrogen atmosphere from 300 to 373 K. The collected data of $\kappa$ from different apparatus (PPMS and LFA) agreed with each other within experimental error at 300 K, validating the reproducible and trustable ultralow $\kappa$ of $AgTlI_2$ bulk sample. Based on the non-preferential orientation of bulk sample (Fig. S3), we therefore regarded it has isotropic thermal transport property and measured $\kappa$ without checking the anisotropic property in our experiments. The nearly identical calculated values of $\kappa_x$ and $\kappa_z$ (see Fig. 4) also suggest that the variation in thermal conductivity among the three Cartesian directions is slight.

## AIMD simulations

A $2 \times 2 \times 2$ supercell with experimental lattice constants at 300 K was used to perform the AIMD simulations with NVT ensemble and obtain the atomic trajectories. An energy cutoff value of 400 eV and a $\Gamma$-centred $1 \times 1 \times 1$ $k$-point mesh were used for AIMD simulations with the PBEsol[52] functional. AIMD trajectory with a time duration of 100 ps was used to visualize the atomic vibration as shown in Fig. S6 and calculate the ADPs. For the calculations of $\sigma^A$ of PbTe, PbSe, AgCl, AgBr, $AgTlI_2$, $Tl_3VSe_4$, InTe, and TlSe at 300 K, we performed AIMD simulations of about 10 ps ($\sigma^A$ has already converged within 10 ps) under the NVT ensemble to obtain the total interatomic forces. For the calculation of $\sigma^A$, harmonic interatomic forces were calculated using the finite displacement method implemented in the Phonopy package[35]. The anharmonic interatomic forces were obtained by subtracting the harmonic forces from the total forces. All specific data reported in Fig. 2b can be found in Table S4.

## Interatomic force constants

Temperature-dependent second-order interatomic force constants (IFCs) were extracted at 300 K using the TDEP[25,26] scheme developed by Hellman et al., as implemented in the hiPhive package[53]. For the unit cell without the consideration of thermal expansion (w/o TE), the lattice constants were relaxed at 0 K ($a = 8.24$ Å; $c = 7.59$ Å). For the unit cell with the thermal expansion (w/ TE), experimental lattice constants ($a = 8.35$ Å; $c = 7.66$ Å) at 300 K were used. 80 configurations in AIMD simulations were randomly selected as the training set. We raised the energy cutoff to 500 eV with a denser $3 \times 3 \times 3$ $k$-point mesh to perform accurate single-point calculations. We obtained the temperature-dependent cubic and quartic IFCs at 300 K using the hiPhive package[53] and our in-house code. The harmonic terms at 0 K calculated using Phonopy[35] were subtracted from the atomic forces and only cubic and quartic force constants were extracted to the residual force-

displacement data. Due to the strong anharmonicity of $AgTlI_2$, we expect that higher-than-fourth-order anharmonic terms may also contribute to the residual atomic forces. Therefore, in the fitting process, we further built the atomic force constant potential up to the sixth-order anharmonicity to fit the residual force-displacement data and excluded the disturbances from the fifth- and sixth-order terms to the cubic and quartic IFCs. We compared the fitting performances of atomic force constant potentials based on the different Taylor expansion truncations up to the fourth- or sixth-order (see Fig. S12) and found that including the fifth- and sixth-order anharmonicity can considerably improve the fitting performance. Note that this fitting scheme was also used by Tadano et al.[54] to reliably extract the third and fourth-order IFCs of strongly anharmonic clathrate $Ba_8Ga_{16}Ge_{30}$.

For $AgTlI_2$, we used the converged neighbour cutoff distances 7.5, 6.0, and 4.2 Å to extract the IFCs of second, third, and fourth-order terms, respectively. For $Ag_8GeS_6$, a $2 \times 2 \times 1$ supercell (240 atoms) was used to perform AIMD simulations at 300 K. We then used the same method to extract the second-, third- and fourth-order IFCs at 300 K with cutoff distances of 7.0, 4.0 and 3.0 Å, respectively.

## Lattice thermal conductivity

We used three different sets of second-order atomic force constants to compute phonon frequency and the group velocity matrix when determining the three different sets of thermal conductivity as shown in (Fig. 4a, b). Three-phonon linewidths were calculated using the ShengBTE[55] package. Four-phonon linewidths were calculated using our in-house code based on the formulae developed by Feng et al.[27,28]. We carefully tested the relation between $q$-point mesh and $\kappa$, and a $9 \times 9 \times 9$ $q$-point mesh was used to compute the phonon linewidths and the $\kappa$ of $AgTlI_2$. The scalebroad of the Gaussian function in the ShengBTE package was set to 1 and 0.25 for the calculations of the three- and four-phonon scatterings, respectively. For $Ag_8GeS_6$, a $6 \times 4 \times 3$ $q$-point mesh was used to compute the $\kappa$, and the scalebroad was set to 1 and 0.1 for three- and four-phonon scatterings, respectively.

## Data availability

All necessary source data files generated for this study are available in the SI repository https://github.com/ZengZezhu/Thermal-conductivity-AgTlI2[72].

## Code availability

The AIMD simulations were performed using the VASP package[30,31]. hiPhive package[53] was used to extract the interatomic force constants. The phonon linewidth and thermal conductivity were computed using the ShengBTE[55] package.

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

## Acknowledgements

We thank Bingqing Cheng (IST Austria) and Terumasa Tadano (NIMS Japan) for reading the manuscript and providing insightful comments. This work is supported by the Research Grants Council of Hong Kong (C7002-22Y and 17318122). ZZ acknowledges the European Union's Horizon 2020 research and innovation programme under the Marie Skłodowska-Curie grant agreement No. 101034413. XS acknowledges funding from the European Union's Horizon 2020 research and innovation programme under the Marie Sklodowska-Curie grant agreement No. 101034329, and the WINNING Normandy Programme supported by the Normandy Region. The computations were performed using research computing facilities offered by Information Technology Services, at the University of Hong Kong.

## Author contributions

Z.Z. conceived the idea; Z.Z. and Y.C. designed the research; Z.Z. wrote the code; Z.Z., R.C. and N.O. performed the calculations; X.S., O.P., P.L., B.R. and E.G. performed the experiments. Z.Z., X.S., O.P., P.L., B.R., E.G., Z.F. and Y.C. wrote the paper.

## Competing interests

The authors declare no competing interests.
