## [Peer Review File · Nature Communications]

Pushing thermal conductivity to its lower limit in crystals with simple structuresREVIEWER COMMENTS

Reviewer #1 (Remarks to the Author):

The manuscript entitled 'Pushing thermal conductivity to its lower limit in crystals with simple structures' delivers a combined experimental and theoretical effort to establish the thermal conductivity of AgTlI₂ and generalize the findings to a batch of low thermal conductive samples. The authors discuss about the ultra-low thermal conductivity of AgTlI₂ and try to model using different theoretical handles like the Finite Displacement Method, the Temperature-dependent Effective Potential, etc. considering both with and without thermal expansion. They show a great match of data with the theoretical models not only for their sample but extends to a generalized low-thermal conductive complex crystal structured sample as well. They come up with a very good observation of the contribution of diffusive thermal transport dominating over specular transport in AgTlI₂ along with many other materials, which, usually in thermoelectrics, is ignored. Overall, this manuscript is of good quality and fulfils the high expectations of the 'Nature Communication' standard. However, I have some small suggestions. I advise this manuscript to be published after those inputs.

1. AgTlI₂ consists of 1D chains. The sintering process can generate an orientational preference for the sample. A powder X-ray diffraction can be better for understanding the actual direction of the thermal conductivity measurement. I suggest the authors check the directional dependence of thermal conductivity, if there is any preferred orientation caused by sintering.
2. There are many materials known to have higher thermal vibrations and displacement parameters causing rattling-like features due to the weak bonding and expressed 6s² lone-pair of Thallium(I) ion. Some of them are cited by the authors as well. But the analysis here says Silver, and not Thallium causes such rattling-like behaviour. Also, the role of Thallium is not very clear in lowering thermal conductivity. Can the authors add a short note on that?
3. In supporting figure S6, the AIMD simulations reveal U11 of TI to have non-linear increment over the temperature range. However other directions and other ions have linear responses. What is the possible origin of this?
4. It will be better if the authors comment on the possible reason for fluctuations in σ in the picosecond time-scale mentioned in figure 2c, keeping in mind, that AgTlI₂ is neither poorly stable, nor superionic in nature?
5. A very minor correction: better if the author provides crystallographic axes in the crystal structure image in figure 1a, b and c.

Reviewer #2 (Remarks to the Author):

Review comments attached (review.pdf)

Reviewer #3 (Remarks to the Author):

The authors have investigated the thermal and structural properties of AgTlI₂, with a specific focus on the compound's thermal transport properties.

The manuscript is commendably well-structured and organized, effectively conveying clear messages. One noteworthy strength of this work lies in its integration of theoretical calculations with experimental data. The inclusion of thermal conductivity measurements is

particularly notable, as such investigations are relatively uncommon in current scientific literature. This experimental approach significantly elevates the quality of the research.

I have particularly enjoyed the comprehensive classification of experimental measurements for low-thermal conductivity compounds, which proof the authors' meticulous engagement with the existing literature in the field. For these reasons, I believe that this paper holds significant value for future research in material science and I recommend its publication. However, I have some concerns that require clarification and resolution prior to its final acceptance.

1) You present both the analysis of the ADP and of the σ^A proposed by Knoop et al. [1]. If I have understood correctly, both these values quantify lattice anharmonicity and consist of a variance computed over an MD trajectory. Do you have any insight on how these two quantities might be related? σ^A clearly is summed over all the atoms, but it might be informative to relate the ADP to an "atomic resolved" σ^A_i , to see if they actually carry different physical information on anharmonicity.

2) I think you should move Fig S9 to the main text, for it bears significant information on your thermal conductivity calculation. In fact, to compute lattice thermal conductivity, you rely on the Wigner transport equation developed by Simoncelli et al. [2,3]. This expression relies on a certain approximation of the phonon self-energy (namely the Lorentzian spectral function approximation [4]) and it is valid when the phonons are well-defined quasiparticles. From the study of the linewidths vs phonon frequencies you present on S9, it appears clear that most of your phonon modes are overdamped (fig 7 of [3], fig 2 of [4]), i.e. possess a lifetime shorter than their period. In this regime, phonons are not well resolved and the Wigner transport equation is not justified. In this scenario, one should resort to a full spectral density calculation of thermal conductivity as the one implemented in [5]. This consists in my major concern with your manuscript: I propose a discussion on the validity of the thermal conductivity formula used in your paper, particularly in light of the observed phonon behavior. This discussion would contribute to solidifying the foundation of your simulation results.

3) Can you give more details on how you compute the phonon velocity matrix? Is it obtained from renormalized phonon frequencies? Does it bear some kind of anharmonic or temperature dependence itself? You can add some details in the methods section.

4) From your results, it seems that calculating thermal conductivity without thermal expansion and finite displacements gives the same result as for the more refined method including thermal expansion and temperature-dependent effective potential. Do you have any insights on this effect? Is it possible that some cancellation of errors or compensation is happening?

5) I get the idea of the sketch you present on Fig. 4 but I have to say that I find it a bit confusing. At first glance, it looks quite busy and I had a hard time interpreting it. I still have some perplexities: what should the blue area represent? Why does it possess this particular shape? As a general comment, I advise against changing your definitions midway, for it results hard to follow. In the introduction, you dubbed "simple" the crystals which feature low off-diagonal thermal conductivity (I agree on this definition). However, in the discussion of Fig. 4 "simple" are now crystals with a small number of atoms in the unit cell. Please unify the nomenclature to avoid confusion in a field already full of ambiguities...

[1]- F. Knoop et al., Physical Review Materials 4, 083809 (2020)

[2]- M. Simoncelli et al., Nature Physics 15, 809–813 (2019)

[3]- M. Simoncelli et al., Phys. Rev. X 12, 041011 (2022)

[4]- G. Caldarelli et al., Phys. Rev. B 106, 024312 (2022)

[5]- D. Dangic et al., npj Computational Materials 7, 57 (2021)

This manuscript measures the thermal conductivity of AgTlI_2 (I4/mcm, InTe structure type) and studies its thermal transport properties with the theory based on the Wigner transport equation. It proposes a path towards the finding of high-performance thermoelectric materials with simple structures. It discusses the need of reaching a compromise between reducing the propagative contribution to the lattice thermal conductivity and increasing the diffusive one. This translates into a compromise between the design of simple structures with few atoms per cell (to limit the increase in thermal conductivity due to wave tunneling between different phonon branches) and the increase in material complexity (to obtain a lower propagative conductivity).

The discussion on the calculated temperature-dependent thermal properties of the material is based on sufficient data and is reliable.

The quality of this paper is high, but I cannot say that it made a breakthrough progress from the viewpoint of thermoelectric materials design. This work certainly deserves to be considered for publication, but it may not satisfy the criteria of Nature Communications for the following reasons:

(1) The computational method adopted in this work is not new. It has already been published in Refs. 13, 15 and 54. The current work can be regarded as an application of the existing methods.

(2)

a. The increase of diffusive thermal conductivity with the complexity of the crystal structure and the decrease of propagative thermal conductivity with increasing anharmonicity are phenomena already known to the community. Trying to enhance the rattling motion in cage-like structure has been one of the main procedures followed for improving the figure of merit of thermoelectric materials.

Do the results of this work on AgTlI_2 provide new understanding for the thermoelectric community?

b. Or, is there a novel strategy to further lower the thermal conductivity or improve the figure of merit?

c. The answers seem to be “no”, at least not in the present manuscript.

Here are my comments:

(1) The authors cite the experimental work (Schiraldi et al.) where electrical transport properties were measured, showing that the silver ion conductor AgTlI_2 exhibit an extremely low electrical conductivity (σ) values (see Figure below). This potentially makes its thermoelectric figure of merit extremely low, and not relevant for thermoelectric applications.

Electrical transport measurements and validation through first-principles simulation may be required to discuss the reliability of AgTlI_2 as a candidate high-performance thermoelectric material.

(2) The authors do not provide the electronic band structure and density of states obtained with the PBEsol functional they used to compute interatomic force constants AgTlI_2 . It should be checked that the ground state of this system is insulating, as expected from the experiments. I would suggest to add this to the discussion in the Supplementary Information.

Although the authors have identified AgTlI_2 as a simple crystal with low thermal conductivity and validate the results with AIMD and first-principles simulations, however, the significance of this work did not meet the Nature Communications standard.

Response letter to reviewers

We express our gratitude to the reviewers for their meticulous review of our manuscript. We highly value the thoughtful concerns, comments, and suggestions provided by the reviewers, which have significantly improved our manuscript.

In response to the reviewers' feedback, we have implemented several changes, and these modifications are indicated in blue in the revised version of the manuscript. The following section addresses each of the points raised by the reviewers.

Reviewer #1 (Remarks to the Author):

The manuscript entitled 'Pushing thermal conductivity to its lower limit in crystals with simple structures' delivers a combined experimental and theoretical effort to establish the thermal conductivity of AgTlI_2 and generalize the findings to a batch of low thermal conductive samples. The authors discuss the ultra-low thermal conductivity of AgTlI_2 and try to model using different theoretical handles like the Finite Displacement Method, the Temperature-dependent Effective Potential, etc. considering both with and without thermal expansion. They show a great match of data with the theoretical models not only for their sample but extends to a generalized low-thermal conductive complex crystal structured sample as well. They come up with a very good observation of the contribution of diffusive thermal transport dominating over specular transport in AgTlI_2 along with many other materials, which, usually in thermoelectrics, is ignored. Overall, this manuscript is of good quality and fulfills the high expectations of the 'Nature Communication' standard. However, I have some small suggestions. I advise this manuscript to be published after those inputs.

We appreciate the reviewer for examining our work and providing a positive evaluation of our work.

AgTlI_2 consists of 1D chains. The sintering process can generate an orientational preference for the sample. A powder X-ray diffraction can be better for understanding the actual direction of the thermal conductivity measurement. I suggest the authors check the directional dependence of thermal conductivity, if there is any preferred orientation caused by sintering.

We thank the reviewer for this comment. XRD measurements on both SPS-ed powder and bulk samples were conducted and presented in Fig. S3 of the Supplementary Information (SI). No significant difference was observed in the XRD diffraction patterns. In the manuscript (see *Thermal transport measurements* in the Methods section), we have discussed the non-preferential orientation of the bulk sample by comparing it with the powder samples. Due to comparable patterns and non-preferential orientation, we treated the bulk sample as exhibiting isotropic thermal transport properties, measuring κ without explicitly checking for anisotropy in our experiments. Calculated results for thermal conductivity at x and z directions have no apparent difference, further verifying the isotropic thermal transport property in AgTlI_2 .

There are many materials known to have higher thermal vibrations and displacement parameters causing rattling-like features due to the weak bonding and expressed 6s² lone-pair of Thallium(Tl) ion. Some of them are cited by the authors as well. But the analysis here says Silver, and not Thallium causes such rattling-like behavior. Also, the role of Thallium is not very clear in lowering thermal conductivity. Can the authors add a short note on that?

We thank the reviewer for this helpful comment. Thallium (Tl), owing to its heavy mass, typically contributes to the low thermal conductivity of materials. This effect arises from its weakened bonding with other atoms, resulting in relatively large thermal displacements (rattling behavior) compared to other elements. However, in AgTlI₂, both our AIMD simulations and experimental measurements reveal that Ag ions exhibit larger atomic displacement parameters (ADP) than Tl ions at room temperature.

To further investigate, we have compiled existing results for the ADP of Tl element in some strongly anharmonic materials with rattling behavior, presenting them in Fig. R1. The maximum ADP of Tl in AgTlI₂ is comparable, and even slightly larger, than the corresponding values observed in these materials that exhibit Tl-rattling behavior (i.e., TlSe, Tl₃VSe₄ and TlInTe₂). This observation prompts us to regard Tl also as a potential rattler in AgTlI₂. Interestingly, in AgTlI₂ and other complex compounds such as Tl₂Ag_{11.86}Se₇, Tl₃Ag₃Sb₂S₆, and Tl₃Ag₃As₂S₆, Ag ions all exhibit stronger rattling vibrations (larger ADPs) than Tl ions at 300 K. Consequently, Ag can be viewed as a more prominent rattler in AgTlI₂, prompting a more detailed discussion of Ag vibrations and their effects on the thermal transport of AgTlI₂ in this work.

From the phonon density of states, Tl atoms dominate the vibrations in the phonon frequency ranges of approximately 0.5~0.8 and 1.0~1.5 THz, crucial for scattering low-frequency phonons and decreasing propagative thermal conductivity. Another influence of Tl atoms on thermal conductivity is the phonon blocking phenomenon, as discussed in our previous work (Phys. Rev. B 106, 054302) for TlXSe₂ (X = Tl, In, or Al) chain-like materials. The stillness of Tl atoms in high-frequency optical phonon modes leads to pronounced in-plane phonon blocking, confining heat flux, and effectively impeding thermal transport. Since AgTlI₂ shares the same chain-like crystal structure with TlXSe₂, and based on the phonon density of states of Tl atoms, we infer a similar phonon blocking phenomenon affecting propagative thermal conductivity along in-plane directions. This is consistent with the spectral thermal conductivity results shown in Fig. 4c-d of the revised manuscript, where the contribution of phonons with frequencies higher than 1.5 THz to propagative thermal conductivity is minimal.

In the revised manuscript, we have included short notes to discuss the relatively large ADP of Tl and its effects on the thermal transport of AgTlI₂. Fig. R1 has also been added to the SI.

Fig. R1 The maximum anisotropic displacement parameters of Tl and Ag atoms at 300 K in some solid materials with typical rattling behavior.

The provided data in Fig. R1 was sourced from the following references:

Tl₂Ag_{11.86}Se₇ (Chem. Mater. 2017, 29, 21, 9565–9571)

TlSe (J. Am. Chem. Soc. 2019, 141, 51, 20293–20299)

TlInTe₂ (Angew. Chem. Int. Ed. 2021, 60, 4259–4265)

TlCuHfSe₃ (Chem. Mater. 2019, 31, 21, 8734–8741)

Tl₂SnTe₅ (J. Solid State Chem. 1999 146, 2, 528–532)

Tl₃Ag₃Sb₂S₆ and Tl₃Ag₃As₂S₆ (J. Alloys Compd. 2008, 457, 1–2, 12)

3. In supporting figure S6, the AIMD simulations reveal U_{11} of Ag to have non-linear increment over the temperature range. However other directions and other ions have linear responses. What is the possible origin of this?

We thank the referee for highlighting this point. Our AIMD simulations reveal that U_{11} , U_{22} and U_{33} of Ag all exhibit a subtle nonlinear increment with increasing temperature. Notably, this phenomenon is not observed in Tl and I elements.

To provide a theoretical context, let us consider a one-dimensional harmonic oscillator, where the total energy (E) of the particle is given by:

$$E = \frac{\hat{p}^2}{2m} + \frac{1}{2}k\hat{x}^2 = \frac{\hat{p}^2}{2m} + \frac{1}{2}m\omega^2\hat{x}^2$$

Here, m is the particle mass, k is the harmonic force constant, $\omega = (k/m)^{1/2}$ is the angular frequency of the oscillator, and x is the atomic position. In classical MD simulations, E is proportional to $k_B T$, where T is the system temperature. Consequently, we observe that the simulated temperature in the system is proportional to the square of the atomic position x . Therefore, it is well-established from many previous studies (such as Tl_3VSe_4 (Phys. Rev. Lett. 2020, 124, 065901) and TlInTe_2 (Angew. Chem. Int. Ed. 2021, 60, 4259–4265)) that the anisotropic displacement parameter (ADP, unit in \AA^2) or mean squared displacement (MSD, unit in \AA^2) enhance linearly with increasing temperature if the lattice anharmonicity of these materials can be treated as a perturbation of the harmonic Hamiltonian. However, when the lattice anharmonicity is strong and cannot be well treated as a perturbation, the nonlinear behavior of ADP (or MSD) occurs naturally. In AgTlI_2 , Ag apparently exhibits stronger anharmonicity compared to Tl and I elements, leading to a slight nonlinear behavior of its ADP with varying temperature.

4. It would be better if the authors comment on the possible reason for fluctuations in σ in the picosecond time-scale mentioned in figure 2c, keeping in mind that AgTlI_2 is neither poorly stable, nor superionic in nature?

We thank the reviewer for this comment. The observed fluctuation is primarily attributed to the substantial thermal vibrations of Ag atoms. In Fig. R2, we present the probability distribution of Ag, Tl, and I atoms in the x - y plane, as computed from the AIMD trajectory at 300 K. Notably, Ag exhibits a broader probability distribution of atomic displacements compared to Tl and I elements.

Upon further analysis, we find a correlation between the thermal displacement of Ag atoms and the instantaneous σ values. In instances with high σ values, Ag atoms undergo large thermal displacement, while in snapshots with relatively low σ values, Ag atoms only slightly deviate from their lattice equilibrium positions. This underscores that the large thermal displacement of Ag is a key factor contributing to the fluctuation of the anharmonic parameter σ .

It is crucial to note that, despite these fluctuations, the equilibrium position of Ag remains well-defined and situated on the lattice site. This observation signifies the stability of the crystal structure of AgTlI_2 at 300 K, with no indication of superionic behavior.

Fig. R2 Atomic displacement probability distribution of Ag, Tl and I atoms in AgTlI_2 computed from AIMD trajectory at 300 K.

5. A very minor correction: better if the author provides crystallographic axes in the crystal structure image in figure 1a, b and c.

We thank the reviewer for noticing this issue. We have included the information of crystallographic axes in Fig. 1 of the revised manuscript.

Reviewer #2 (Remarks to the Author):

This manuscript measures the thermal conductivity of AgTlI_2 (I4/mcm, InTe structure type) and studies its thermal transport properties with the theory based on the Wigner transport equation. It proposes a path towards the finding of high-performance thermoelectric materials with simple structures. It discusses the need of reaching a compromise between reducing the propagative contribution to the lattice thermal conductivity and increasing the diffusive one. This translates into a compromise between the design of simple structures with few atoms per cell (to limit the increase in thermal conductivity due to wave tunneling between different phonon branches) and the increase in material complexity (to obtain a lower propagative conductivity).

We appreciate the reviewer for the meticulous review of our work. We would like to emphasize that the primary contribution of this study lies in proposing a pathway to identify simple crystals with low thermal conductivity, using AgTlI_2 as an example. We appreciate the reviewer's summary of our proposed pathway for achieving low thermal conductivity in crystals with simple structures. However, this work indeed does not present a definitive method for discovering high-performance thermoelectric materials. We acknowledge that the last paragraph of our initial manuscript (about the thermal conductivity of complex thermoelectric materials) may have led to confusion, and we have revised it in the updated manuscript to provide a more accurate presentation of our research focus.

We have also explicitly stated in the revised manuscript that:

'Given that AgTlI_2 is an insulator with a large electronic band gap (Fig. S11), it is not suitable for direct thermoelectric applications without proper alloying or doping.'

The discussion on the calculated temperature-dependent thermal properties of the material is based on sufficient data and is reliable.

We thank the reviewer for the positive evaluation of our calculations in this work.

The quality of this paper is high, but I cannot say that it made a breakthrough progress from the viewpoint of thermoelectric materials design. This work certainly deserves to be considered for publication, but it may not satisfy the criteria of Nature Communications for the following reasons:

We appreciate the positive assessment of our work's quality by the reviewer. We would like to highlight the contributions of this work to the materials science community:

- Our comprehensive approach, combining advanced experiments and calculations, focuses on discovering simple crystals with ultralow thermal conductivity—a challenging endeavor not extensively explored in previous studies.
- The identification of AgTlI_2 , defying conventional expectations with an exceptionally low thermal conductivity of 0.25 W/mK at room temperature, challenges the notion that complex materials always possess lower thermal conductivity.
- This finding represents a breakthrough in pushing the thermal conductivity of crystals with simple crystal structures to its lower limit. Importantly, the insights gained into thermal transport mechanisms in this work provide an alternative pathway for achieving ultralow thermal conductivity in simple crystals.

The computational method adopted in this work is not new. It has already been published in Refs. 13, 15 and 54. The current work can be regarded as an application of the existing methods.

We appreciate the reviewer's observation regarding the computational method employed in our work. While our focus is not to propose a new methodology for assessing thermal conductivity, we would like to emphasize that the approach utilized in this work represents a state-of-the-art method in lattice dynamics, particularly tailored for addressing the thermal transport of strongly anharmonic materials.

The increase of diffusive thermal conductivity with the complexity of the crystal structure and the decrease of propagative thermal conductivity with increasing anharmonicity are phenomena already known to the community. Trying to enhance the rattling motion in cage-like structure has been one of the main procedures followed for improving the figure of merit of thermoelectric materials.

- Do the results of this work on AgTlI_2 provide new understanding for the thermoelectric community?
- Or, is there a novel strategy to further lower the thermal conductivity or improve the figure of merit?
- The answers seem to be “no”, at least not in the present manuscript.

It is noted that the focus of this work is to provide an alternative approach to push thermal conductivity to its lower limit in crystals with simple structures. The design of high-performance thermoelectric materials requires more complex engineering of both thermal and electrical transport properties, which is beyond the scope of the current work.

The authors cite the experimental work (Schiraldi et al.) where electrical transport properties were measured, showing that the silver ion conductor AgTlI_2 exhibit an extremely low electrical conductivity (σ) values (see Figure below). This potentially makes its thermoelectric figure of merit extremely low, and not relevant for thermoelectric applications. Electrical transport measurements and validation through first-principles simulation may be required to discuss the reliability of AgTlI_2 as a candidate high-performance thermoelectric material.

We thank the reviewer for this suggestion. Based on both prior experimental measurements and our first-principles calculations (see Fig. R3), AgTlI_2 is recognized as an insulator. In this study, our primary focus is on minimizing thermal conductivity in crystals with simple structures rather than on its electrical transport properties (or thermoelectric applications). Therefore, we believe the measurements of electrical transport properties are not directly aligned with the primary objectives of this work.

We acknowledge the importance of electrical transport properties in evaluating thermoelectric materials and recognize that further experimental measurements and first-principles calculations may be valuable for a comprehensive assessment. However, given the focus of this work, we contend that the low electrical conductivity of AgTlI_2 , as mentioned in the cited work by Schiraldi et al., does not impede the conclusions drawn from our study.

The authors do not provide the electronic band structure and density of states obtained with the PBEsol functional they used to compute interatomic force constants AgTlI_2 . It should be checked that the ground state of this system is insulating, as expected from the experiments. I would suggest to add this to the discussion in the Supplementary Information.

We thank the reviewer for this comment. We computed the electrical band structure and density of states of AgTlI_2 using the PBEsol functional, and the results are showed in Fig. R3. The band gap is 1.23 eV, indicating the AgTlI_2 is an insulator. We have discussed the results in the last paragraph of the revised manuscript and moved Fig. R3 into the Supplementary Information.

Fig. R3 The electrical band structure and density of states of AgTlI_2 calculated using the VASP package with PBEsol functional.

Although the authors have identified AgTlI_2 as a simple crystal with low thermal conductivity and validate the results with AIMD and first-principles simulations, however, the significance of this work did not meet the Nature Communications standard.

We thank the reviewer for the positive evaluation of our work, and we hope our explanations and revisions can more clearly highlight the contributions and significance of this work and address the concerns of the reviewer.

Reviewer #3 (Remarks to the Author):

The authors have investigated the thermal and structural properties of AgTlI₂, with a specific focus on the compound's thermal transport properties.

The manuscript is commendably well-structured and organized, effectively conveying clear messages. One noteworthy strength of this work lies in its integration of theoretical calculations with experimental data. The inclusion of thermal conductivity measurements is particularly notable, as such investigations are relatively uncommon in current scientific literature. This experimental approach significantly elevates the quality of the research.

I have particularly enjoyed the comprehensive classification of experimental measurements for low-thermal conductivity compounds, which proof the authors' meticulous engagement with the existing literature in the field. For these reasons, I believe that this paper holds significant value for future research in material science and I recommend its publication. However, I have some concerns that require clarification and resolution prior to its final acceptance.

We are very grateful to the reviewer for the positive assessment and recommendation regarding our work.

You present both the analysis of the ADP and of the σ^A proposed by Knoop et al. [1]. If I have understood correctly, both these values quantify lattice anharmonicity and consist of a variance computed over an MD trajectory. Do you have any insight on how these two quantities might be related? σ^A clearly is summed over all the atoms, but it might be informative to relate the ADP to an "atomic resolved" σ^{Ai} , to see if they actually carry different physical information on anharmonicity.

We appreciate the reviewer for this insightful comment. The relation between ADP (or atomic thermal displacement) and anharmonic parameter σ^A may be comprehended by a simple analytical model. Considering a monatomic system where its potential energy U , total atomic force F , anharmonic atomic force F^A and anharmonic parameter σ^A can be respectively written as a function of atomic displacement x as

$$U = \frac{1}{2}k_0x^2 + \frac{1}{3}k_1x^3,$$
$$F = \frac{-\partial U}{\partial x} = -(k_0x + k_1x^2),$$
$$F^A = -k_1x^2,$$
$$\sigma^A = \frac{k_1x^2}{k_1x^2 + k_0x} = \frac{k_1}{k_1 + \frac{k_0}{x}}.$$

It is evident that the anharmonic parameter will be increased with the enhancement of atomic displacement. Considering the ADP (or mean square displacement) is linearly proportional to the square of atomic displacement in most materials with relatively modest lattice anharmonicity, we can conclude that for this analytical system, the σ^A is positively related to the ADP.

However, for some strongly anharmonic materials with complex potential energy surfaces, such as the U-type energy surface of PbTe (Phys. Rev. B 91, 214310) and the double-well energy surface of Cu₁₂Sb₄S₁₃ (Phys. Rev. Lett. 125, 085901), due to the involvement of high-order lattice anharmonicity, there is no clear and concise relation between ADP and anharmonic parameter σ^A , as the potential energy cannot be expressed as a simple cubic polynomial with respect to the atomic displacement.

In Fig. 2b of the revised manuscript, we see that there is a large deviation from the power-law model for σ^A of Tl₃VSe₄ as the σ^A is small while its thermal conductivity is very low, although the MSD of Tl in Tl₃VSe₄ is larger than that of Pb in PbTe at 300 K. This can be understood by the mode-resolved degree of anharmonicity measure (Phys. Rev. Materials 4, 083809) in Tl₃VSe₄. Due to the existence of light element V, there are some high-frequency phonon modes dominated by V atoms that represent typical harmonic vibrational behavior and thus small mode-resolved σ^A . While low-frequency phonon modes dominated by Tl atoms have strong anharmonicity and thus large mode-resolved σ^A . However, the anharmonic parameter σ^A represents the average of the system's anharmonicity and it intrinsically includes all phonon mode vibrations. Therefore, although Tl atoms have a large atomic thermal displacement in Tl₃VSe₄, the anharmonic parameter σ^A is not so large at 300 K.

We have incorporated the suggestions of the reviewer and calculated the atomic-resolved anharmonic parameters for AgTlI₂, as illustrated in Fig. R4. It is evident that Ag ions exhibit a larger σ^A (indicating stronger anharmonicity) compared to Tl and I atoms. Moreover, the fluctuation in the anharmonic parameter σ^A of AgTlI₂ is significantly influenced by the fluctuation in σ^A of Ag ions, whereas Tl and I display only subtle fluctuations. These observations further underscore the crucial role of the rattling behavior of Ag ions in AgTlI₂ in determining its thermal transport.

Fig. R4 Atomic-resolved anharmonic parameters for Ag, Tl and I elements in AgTlI₂ calculated using the AIMD trajectory at 300 K.

I think you should move Fig S9 to the main text, for it bears significant information on your thermal conductivity calculation.

We thank the reviewer for this suggestion. We have moved Fig. S9 to the revised manuscript as Fig. 3.

In fact, to compute lattice thermal conductivity, you rely on the Wigner transport equation developed by Simoncelli et al.[2,3]. This expression relies on a certain approximation of the phonon self-energy (namely the Lorentzian spectral function approximation [4]) and it is valid when the phonons are well-defined quasiparticles. From the study of the linewidths vs phonon frequencies you present on S9, it appears clear that most of your phonon modes are overdamped (fig 7 of [3], fig 2 of [4]), i.e. possess a lifetime shorter than their period. In this regime, phonons are not well resolved and the Wigner transport equation is not justified. In this scenario, one should resort to a full spectral density calculation of thermal conductivity as the one implemented in [5]. This consists in my major concern with your manuscript: I propose a discussion on the validity of the thermal conductivity formula used in your paper, particularly in light of the observed phonon behavior. This discussion would contribute to solidifying the foundation of your simulation results.

[1]- F. Knoop et al., Physical Review Materials 4, 083809 (2020)

[2]- M. Simoncelli et al., Nature Physics 15, 809–813 (2019)

[3]- M. Simoncelli et al., Phys. Rev. X 12, 041011 (2022)

[4]- G. Caldarelli et al., Phys. Rev. B 106, 024312 (2022)

[5]- D. Dangi et al., npj Computational Materials 7, 57 (2021)

We appreciate the reviewer for pointing out this insightful issue. In our work, all presented phonon frequencies follow the convention of linear frequency f , computed from the angular frequency ω as $f = \omega/2\pi$, aligning with the Phonopy package standard. However, in the output file (BTE.w_anharmonic) generated by the ShengBTE package for plotting Fig. S9, the phonon frequencies are in angular units. To ensure consistency, we divided the angular frequency by 2π when presenting it. Unfortunately, this adjustment was not applied to the phonon linewidth, leading to an inadvertent discrepancy. We apologize for any confusion caused and have rectified our oversight. The corrected representation is included in Fig. 3 of the revised manuscript. Now we see that there is no distinct overdamped phonon behavior in AgTlI₂ at 300 K. Note that the plotting of the phonon linewidths of Ag₈GeS₆ has also been revised accordingly.

We carefully have studied Refs. [4] and [5] as the reviewer raised here. Based on the derivation of thermal conductivity using the Wigner definition of the heat flux operator in Ref. [5], if the phonon spectral densities have the Lorentzian line shape (well-defined phonons), we can derive the two-channel thermal conductivity as we wrote in Eq. 2 of our manuscript. Dangi et al. (Ref. [4]) also proposed the two-channel thermal conductivity based on the Hardy's heat flux. In practice, these two methods only have a subtle difference for the computed thermal conductivity at high temperatures and the difference can usually be ignored as demonstrated in Ref. [5] by Caldarelli et al.. For ill-defined phonons with non-Lorentzian spectral shape, Refs. [4] and [5] both proposed an integral of phonon spectral function to rigorously access the thermal conductivity, although the definition of group velocity matrix is slightly different.

Regarding AgTlI₂, as observed in Fig. 3 of the revised manuscript, only a very small amount of low-frequency phonons exhibit linewidths slightly exceeding their frequencies when using FDM with thermal expansion to

compute phonon properties. If we compute the phonons properties using TDEP method with thermal expansion, all phonons are well defined. This observation supports the viability of the unified theory employing the Lorentzian spectral function approximation as we used in this work for thermal conductivity assessment. Furthermore, our experimental findings also align with these calculations.

We fully endorse the reviewer's suggestion regarding a comprehensive discussion on the applicable regime of the widely adopted unified theory proposed in 2019 by Simoncelli et al.. We have incorporated additional insights and references (Refs. [4] and [5]) in the revised manuscript to provide readers with a more thorough information.

Can you give more details on how you compute the phonon velocity matrix? Is it obtained from renormalized phonon frequencies? Does it bear some kind of anharmonic or temperature dependence itself? You can add some details in the methods section.

We used the same second-order force constants to compute both the corresponding phonon frequency and group velocity matrix in the calculations of the thermal conductivity. For example, the temperature-dependent second-order force constants were used to compute both temperature-dependent phonon frequency and group velocity matrix, which intrinsically include the anharmonic corrections.

We thank the reviewer for this suggestion, and we have added this information to the Method section of the revised manuscript.

From your results, it seems that calculating thermal conductivity without thermal expansion and finite displacements gives the same result as for the more refined method including thermal expansion and temperature-dependent effective potential. Do you have any insights on this effect? Is it possible that some cancellation of errors or compensation is happening?

We thank the reviewer for this comment. We concur with the observation that the good agreement between the computed thermal conductivity (using FDM without TE) and the experimental results can be attributed to the cancellation of computational errors.

The lattice thermal expansion, in a broad sense, induces phonon softening, leading to two primary effects: i) a reduction in phonon group velocity and ii) an enhancement of low-frequency phonon scattering (due to stronger coupling of acoustic and optical phonons). These changes contribute to the decrease in thermal conductivity. Consequently, neglecting thermal expansion tends to result in an overestimation of the thermal conductivity.

Additionally, previous investigations have demonstrated the significance of anharmonic phonon frequency renormalization in strongly anharmonic materials such as Ti_3VSe_4 (Phys. Rev. Lett. 124, 065901), $\text{Cs}_2\text{PbI}_2\text{Cl}_2$ (Phys. Rev. B 105, 184303) and SrTiO_3 (Phys. Rev. B 104, 235205). In these cases, the anharmonic correction plays a crucial role in enhancing low-frequency phonon frequencies, ultimately leading to an increase in the computed thermal conductivity. In other words, utilizing phonon frequencies computed from the finite displacement method (without anharmonic correction and phonon hardening) can underestimate the thermal conductivity.

Therefore, using FDM without considering TE can lead to error cancellation, contributing to the observed agreement between the computed and the experimental thermal conductivities.

Interestingly, Lindsay et al. (Science 360, 1455–1458 (2018)) utilized the FDM to compute the thermal conductivity of Ti_3VSe_4 , validating the two-channel model they proposed. Their simulated results align well with the experiments. In contrast, Xia et al. (Phys. Rev. Lett. 124, 065901 (2020)) and Zeng et al. (Phys. Rev. B 103, 224307 (2021)) recalculated the thermal conductivity of Ti_3VSe_4 , incorporating anharmonic corrections to various high-order terms. All these three studies successfully reproduced the experimental results, albeit with different lattice anharmonic corrections.

I get the idea of the sketch you present on Fig. 4 but I have to say that I find it a bit confusing. At first glance, it looks quite busy and I had a hard time interpreting it. I still have some perplexities: what should the blue area represent? Why does it possess this particular shape?

As a general comment, I advise against changing your definitions midway, for it results hard to follow. In the introduction, you dubbed "simple" the crystals which feature low off-diagonal thermal conductivity (I agree on this definition). However, in the discussion of Fig. 4 "simple" are now crystals with a small number of atoms in the unit cell. Please unify the nomenclature to avoid confusion in a field already full of ambiguities...

We appreciate the reviewer for pointing out this issue. We agree with the reviewer that the division of different regions based on different colors is confusing to readers. We have removed the confused information and replot Fig. 4 and show it in Fig. 5 of the revised manuscript (also attached below).

In this revised figure, we introduce a red dashed straight line to illustrate the current protocol for discovering materials with lower thermal conductivity. Additionally, we have updated the x -label from "complexity of unit cell of crystal" to "Total number of atoms in the primitive cell of crystals," providing a clearer and simpler representation and avoiding the confusing definition of terminologies.

Furthermore, we have made revisions by relocating the text " κ_{pg} -dominated" to the upper left and " κ_{diff} -dominated" to the lower right. In the upper left, where there are a few atoms in the unit cell, the bunching of phonon branches is weak, resulting in a small and negligible diffusive thermal conductivity. As the number of atoms in the unit cell increases, severe overlap of phonon spectral functions occurs, leading to a significant rise in diffusive thermal conductivity.

REVIEWERS' COMMENTS

Reviewer #2 (Remarks to the Author):

The authors have responded comprehensively to the comments, the final state of the article is valid for publication.

Reviewer #3 (Remarks to the Author):

I am satisfied with the response of the authors to my concerns about their research. I have no further comments and the paper can proceed to publication.

Response letter to reviewers

Reviewer #2 (Remarks to the Author):

The authors have responded comprehensively to the comments, the final state of the article is valid for publication.

Reviewer #3 (Remarks to the Author):

I am satisfied with the response of the authors to my concerns about their research. I have no further comments and the paper can proceed to publication.

Response:

We highly appreciate the reviewers for their insightful comments and positive support toward the publication of this work!